# Pertussis Toxin Inhibits Encephalitogenic T-Cell Infiltration and Promotes a B-Cell-Driven Disease during Th17-EAE

**DOI:** 10.3390/ijms22062924

**Published:** 2021-03-13

**Authors:** Zahra Maria, Emma Turner, Agnieshka Agasing, Gaurav Kumar, Robert C. Axtell

**Affiliations:** 1Department of Arthritis and Clinical Immunology, Oklahoma Medical Research Foundation, Oklahoma City, OK 73104, USA; zahra-maria@omrf.org (Z.M.); Emma-Turner@okstate.edu (E.T.); a.m.agasing@gmail.com (A.A.); gaurav-kumar@omrf.org (G.K.); 2Department of Animal and Food Sciences, Oklahoma State University, Stillwater, OK 74078, USA

**Keywords:** experimental autoimmune encephalomyelitis, multiple sclerosis, chemokines, Th17, B cells

## Abstract

Pertussis toxin (PTX) is a required co-adjuvant for experimental autoimmune encephalomyelitis (EAE) induced by immunization with myelin antigen. However, PTX’s effects on EAE induced by the transfer of myelin-specific T helper cells is not known. Therefore, we investigated how PTX affects the Th17 transfer EAE model (Th17-EAE). We found that PTX significantly reduced Th17-EAE by inhibiting chemokine-receptor-dependent trafficking of Th17 cells. Strikingly, PTX also promoted the accumulation of B cells in the CNS, suggesting that PTX alters the disease toward a B-cell-dependent pathology. To determine the role of B cells, we compared the effects of PTX on Th17-EAE in wild-type (WT) and B-cell-deficient (µMT) mice. Without PTX treatment, disease severity was equivalent between WT and µMT mice. In contrast, with PTX treatment, the µMT mice had significantly less disease and a reduction in pathogenic Th17 cells in the CNS compared to the WT mice. In conclusion, this study shows that PTX inhibits the migration of pathogenic Th17 cells, while promoting the accumulation of pathogenic B cells in the CNS during Th17-EAE. These data provide useful methodological information for adoptive-transfer Th17-EAE and, furthermore, describe another important experimental system to study the pathogenic mechanisms of B cells in multiple sclerosis.

## 1. Introduction

Multiple sclerosis (MS) is an autoimmune disease of the central nervous system (CNS) characterized by inflammation, demyelination and axonal loss. Although earlier concepts of MS mainly focused on T cells as the inflammatory mediators, the success of anti-CD20 B-cell-depleting therapies demonstrates the pathogenic contribution of B cells as well [1,2,3]. B cells are able to induce neuroinflammation by differentiating into autoantibody-secreting plasma cells, presenting self-antigens to autoreactive T cells and secreting pro-inflammatory cytokines [4,5,6,7]. However, experimental and clinical studies reveal that B cells also have a regulatory role in MS [8,9]. Experimental autoimmune encephalomyelitis (EAE) is a collection of autoimmune demyelinating models that are commonly used to study MS. It is well known that the interplay between activated B and T cells is important in the exacerbation of MS [10,11,12,13]. However, there are few EAE models in which B cells are necessary for the initiation of disease. The most established model is where C57BL/6 mice are immunized with recombinant human myelin oligodendrocyte glycoprotein (rhMOG), [5,14,15]. Therefore, in order to comprehensively understand the diverse roles of B cells during MS, the development of additional B-cell-dependent EAE models is necessary.

Pertussis toxin (PTX) is an exotoxin derived from *Bordetella pertussis* and its functional role during EAE has been a longstanding enigma. PTX is well known for inhibiting the chemokine-dependent recruitment of immune cells to target tissues or sites of inflammation via its ADP-ribosyltransferase activity on G-protein coupled receptors (GPCRs) [16,17,18,19,20,21]. Conversely, in active EAE models, PTX is proven to be a necessary co-adjuvant for clinical disease presentation [22,23]. However, the exact mechanism of PTX adjuvanticity remains unclear. Historically, PTX has been thought to increase the permeability of the blood–brain barrier to give access to the CNS-infiltrating immune cells [24,25]. In the recent literature, PTX adjuvanticity has been attributed to its ability to induce IL-1β secretion by myeloid cells, which is required to prime auto-reactive Th1 and Th17 cells in the peripheral tissues [26,27,28]. Although it has been reported that PTX reduces EAE in the adoptive transfer or passive model of EAE, where myelin-specific T helper cells are directly transferred into a recipient animal, the mechanism behind this is not well elucidated [29,30].

The purpose of this study was to investigate the role of PTX during Th17-EAE and characterize the alterations in disease pathogenesis. In doing so, we made two important observations. First, we found that PTX treatment reduced, but did not block, Th17-EAE disease, by directly inhibiting the migration of pathogenic Th17 cells into the CNS. Second, we found that the neuroinflammation in the PTX-treated Th17-EAE is dependent on an inflammatory B-cell function.

## 2. Results

### 2.1. Pertussis Toxin Reduced Th17-EAE Disease Severity

In order to investigate the role of PTX in the adoptive transfer Th17-EAE model, we administered PTX or vehicle to mice on the day of and two days after the transfer of myelin-specific Th17 cells. Our data show that PTX treatment significantly ameliorated Th17-EAE by reducing disease severity and delaying the onset of paralysis (Figure 1A–C and Appendix A) compared to the vehicle-treated (No PTX) group. We performed immunohistochemistry on brain and spinal cord sections of mice sacrificed at the peak of disease and verified the disease scores with the extent of demyelination. The mice receiving PTX had fewer and smaller demyelinated lesions compared to control mice (Figure 1C and Appendix A). Next, we investigated the effect of PTX on infiltrating immune cell populations (Appendix A) in the brain (Figure 1E) and spinal cord (Figure 1F) of the Th17-EAE mice. Our results show that PTX significantly reduced the accumulation of CD4^+^ T helper cells in the brain and spinal cord. In addition, PTX diminished the number of macrophages in the brain and reduced the number of neutrophils in the spinal cord of mice. In contrast, we observed an increase in B cell numbers in both the brain and spinal cord in the mice treated with PTX (Figure 1E,F). Interestingly, we found that PTX increased the number of MHCII^+^ B cells in the brain (Figure 1E and Appendix A). In addition, we observed an increase in class-switched memory B cells and plasma cell/plasma blast numbers in the spleens of the PTX-treated mice compared to vehicle control mice (Appendix A). Although we did not find any difference in the number of B cells, we saw a significant increase in the MHCII (Appendix A), CD80 (Appendix A) and CD86 (Appendix A) expression (mean fluorescence intensity) on the splenic B cells of PTX-treated mice. These findings show that PTX decreases the accumulation of CD4^+^ T helper cells and myeloid cells in the CNS, thereby delaying disease onset and disease severity. Strikingly, our data also suggest that PTX directly or indirectly promotes the accumulation of B cells in the CNS, and B cells in the PTX-treated mice have an elevated capacity to present antigens to T cells in the periphery.

### 2.2. PTX Reduced Accumulation of Encephalitogenic CD4^+^ T Cells in the CNS

Next, we investigated the effects of PTX on the infiltration of inflammatory CD4^+^ T helper cell populations in the brain and spinal cord in mice with Th17-EAE. We found that PTX significantly decreased the number of CD4^+^IL17A^+^, CD4^+^IFNγ^+^, CD4^+^GM-CSF^+^, CD4^+^IL17A^+^GM-CSF^+^ and CD4^+^IL17^+^IFNγ^+^ cells in the brain and spinal cord of mice with Th17-EAE (Appendix A). We then determined the effects of PTX treatment on the donor Th17 cells by using a congenic adoptive transfer strategy. Here, we used congenic CD45.1^+^ mice to generate Th17 donor cells to transfer into CD45.2^+^ recipient mice and the recipient mice were treated with PTX or PBS on day 0 and day 2 post-transfer (Figure 2A). We found that ~90% of the CD4^+^ T cells in the CNS that expressed IL17A and GM-CSF were CD45.1^+^ donor-derived (data not shown). Furthermore, we found that PTX reduced the absolute number of CD4^+^CD45.1^+^ donor T cells in the brain (Figure 2D) and spinal cord (Figure 2E) of PTX-treated recipient mice. Analysis of the cytokine-producing CD4^+^CD45.1^+^ donor cells revealed that there was a decrease in the absolute number of CD4^+^IL17A^+^ cells in the brain and spinal cord (Figure 2B–E and Appendix A), with an additional reduction in CD4^+^GM-CSF^+^,CD4^+^IL17A^+^GM-CSF^+^ and CD4^+^IFNγ^+^ cells in the brain (Figure 2B,D) and CD4^+^IL17A^+^ IFNγ^+^ in the spinal cord (Figure 2C–E). Since GM-CSF-producing Th17 cells have been reported to be crucial for the induction of EAE [31,32], we speculated that the delayed disease onset and severity could be attributed to the restricted entry of this encephalitogenic T cell subset into the CNS.

### 2.3. PTX Impaired Chemokine-Dependent Recruitment of Inflammatory Th17 Cells

Immune cell trafficking is known to be inhibited by PTX through the desensitization of GPCRs to its ligands [16,17]. In addition, the chemokine receptors CCR2 and CCR6 are important for the homing and recruitment of pathogenic GM-CSF^+^ Th17 cells into the CNS during EAE [33]. Therefore, we speculated that PTX may inhibit the infiltration of the pathogenic T-cell subsets into the CNS by inhibiting CCR2- and CCR6-dependent trafficking. To test this, we conducted an in vitro transmigration assay to determine the effect of PTX on CCL2- and CCL20-induced trafficking of myelin-specific Th17 cells (Figure 3). We found that PTX-treated Th17 cells were less responsive to CCL2- (Figure 3A–C) and CCL20-dependent (Figure 3D) migration. We observed a significant reduction in the migration index of CCL2 and CCL20 of activated CD4^+^IL17A^+^ T helper cells that were cultured in the presence of PTX. In addition, PTX-treated CD4^+^GM-CSF^+^, CD4^+^IFNγ^+^ and CD4^+^IL17^+^IFNγ^+^ T cells were less responsive to CCL2 compared to the control (Figure 3C). Together, these findings indicate that PTX directly inhibits inflammatory CD4^+^IL17A^+^, CD4^+^GM-CSF^+^, CD4^+^IFNγ^+^ and CD4^+^IL17^+^IFNγ^+^ T helper cells from infiltrating the CNS during Th17-EAE.

### 2.4. In Vitro PTX treatment Reduced the Capacity of Th17 Cells to Induce EAE

Our in vitro experiments demonstrate that PTX inhibits the trafficking of encephalitogenic CD4^+^ T cells. We next determined if in vitro PTX treatment of the donor cells inhibits the CNS infiltration of the Th17 cells and reduces disease progression. In order to test this, we polarized encephalitogenic Th17 cells in the presence (in vitro PTX) or absence (No PTX) of PTX before transferring into healthy recipient mice. Prior to transfer, we did not observe any difference in cytokine production between Th17 cells cultured with or without PTX (Appendix A). However, we did observe that mice receiving the in vitro PTX-treated Th17 cells had reduced disease severity, with significantly delayed disease onset of EAE, compared to the control mice (Figure 4A). In addition, there was a significant decrease in the absolute numbers of all CD4^+^ T cells (Figure 4B) as well as IL17A^+^, GM-CSF^+^ and IL17A^+^GM-CSF^+^ CD4^+^T helper cells in both the brain and spinal cord (Figure 4C) of the in vitro PTX group. A significant decrease in the absolute number of IFNγ+ and IL17A^+^IFNγ^+^ CD4^+^ T helper cells was observed in the brain and spinal cord, respectively. These findings support the hypothesis that PTX functionally inhibits the recruitment of IL17A^+^, GM-CSF^+^ and IFNγ^+^ CD4^+^ T helper cells into the CNS. Furthermore, although we did not see a difference in macrophage recruitment, we found a significantly lower number of neutrophils in the brain and spinal cord of the in vitro PTX group (Figure 4D). Since Th17 cells play a role in the recruitment of inflammatory neutrophils during neuroinflammation [34,35], the lower number of neutrophils is likely due to the decreased number of infiltrating T cells rather than PTX itself. Similar to in vivo PTX treatment, we found that the in vitro PTX mice had an increased frequency of MHCII^+^ B cells in the brain compared to control mice (Figure 4E,F and Appendix A). These data show that PTX directly attenuates the trafficking of inflammatory CD4^+^ T helper cells into the CNS, which in turn decreases the infiltration of myeloid cells, but not B cells, and decreases the severity of EAE.

### 2.5. PTX Treatment Reveals a Pro-Inflammatory Function of B Cells in the Th17-EAE Model

B cells can play either an inflammatory or a regulatory function in EAE, which is dependent on the experimental model. We observed that PTX treatment increases the infiltration of B cells in the CNS. Therefore, PTX could be promoting a regulatory function of B cells that reduces disease. Alternatively, B cells may have an inflammatory function that allows disease to occur after PTX treatment. To resolve this question, we compared the effects of PTX on Th17-EAE in wild-type mice with B-cell-deficient µMT mice. We found that, without PTX treatment, the severity of Th17-EAE was equivalent in wild-type mice and the µMT mice. In contrast, with PTX treatment, the µMT mice had significantly less disease compared to the wild-type mice (Figure 5A, Appendix A). In agreement with the clinical scores, we found that the PTX-treated µMT mice had the lowest number of lesions in the brain compared to the other groups; interestingly, there were lesions present in the spinal cords of µMT mice treated with PTX (Figure 5B, Appendix A and Figure 5E). In addition, the PTX-treated µMT mice had significantly fewer total CD4^+^ T cells (Figure 5C,5D) and CD4^+^ T cells expressing IL-17A and GM-CSF in the spinal cord and brains compared to the other groups (Appendix A). These data reveal that B-cell deficiency attenuates paralysis in the PTX-treated mice and, therefore, demonstrate that B cells are critical for promoting demyelination and paralysis during PTX-treated Th17-EAE.

### 2.6. Disease Duration and/or Severity Affects B-Cell Accumulation in the CNS 

The data shown thus far demonstrate that PTX treatment reduces Th17-EAE and promotes a B-cell-dependent disease phenotype. However, it remains unclear which factors are driving the accumulation of inflammatory B cells in the CNS of PTX-treated mice. PTX could be directly stimulating an inflammatory B-cell function in these mice. Alternatively, PTX may induce a delay in disease initiation, which allows the accumulation of inflammatory B cells in the CNS that would naturally infiltrate the CNS at a later stage of the disease [9]. In our experiments, we observed that approximately 50% of the Th17-EAE with no PTX mice reach a severe disease score by day 14–17 that requires euthanasia (Figure 1B); the remaining 50% of the mice had a milder EAE and could be sacrificed at a later time point. To determine if CNS infiltration of B cells was simply a function of time rather than a function of PTX stimulation, we compared B-cell infiltration in mice with severe Th17-EAE that were sacrificed at day 14 with mice that had milder Th17-EAE that were sacrificed at day 22 (Figure 6A). Strikingly, we found that the proportion of B cells in the spinal cords was greater in the mice with milder disease that were sacrificed at day 21 compared to the mice with more severe disease that were sacrificed on day 14. Conversely, the proportion of neutrophils and macrophages was higher in the spinal cords of the mice with severe disease (Figure 6B and Appendix A). The number of CD4^+^ T cells were not different in these two groups of mice. These data suggest that B-cell accumulation in the CNS is directly correlated with the duration of disease, which can only be observed in mice with milder disease.

## 3. Discussion

The key findings of this study are that 1) PTX significantly reduces Th17-EAE by delaying disease onset and reducing disease severity and is protective against CNS demyelination, and 2) B cells play a significant role in disease pathogenesis in this model. PTX, a systemic co-adjuvant, along with MOG/CFA, is necessary for the efficient induction of EAE [22,23]. Early studies speculated that PTX increases the permeability of the blood–brain barrier and allows immune cells to gain access to the CNS [24,25,36,37]. However, PTX may act as an adjuvant by stimulating inflammatory myeloid cells or suppressing CD4^+^CD25^+^ T-regs during the induction phase of EAE [26,27,28,38]. In contrast to its adjuvant function, PTX is well known for its inhibitory effects on homing and recirculating lymphocytes in multiple disease models. PTX has been shown to block chemokine-induced migration and infiltration of immune cells into target organs and sites of inflammation [16,18,19,20,39]. Due to this dual function, the complete role of PTX during EAE remains a conundrum. In contrast to the EAE by the immunization of myelin antigen, where PTX is essential for disease induction, in the adoptive transfer or passive EAE model, it has been shown that donor cells harvested from PLP139-151-immunized mice and cultured in the presence of PTX impart a less severe disease in the recipient mice [29]. Additionally, it has been reported that in-vitro-polarized primary Th17, harvested from MOG-immunized mice, were not able to induce disease in recipient mice which were treated with PTX [30]. However, the mechanism and the immune cell subsets involved in PTX-induced disease suppression remain to be investigated. In order to investigate this paradox, we performed a systematic comparison of the disease phenotype of the Th17-EAE model with and without PTX. Here, we addressed two queries: first, the causation behind the delayed disease onset and reduced disease severity; second, the role of B cells in disease progression in this model.

Our results show that the ameliorated disease severity in PTX-treated Th17-EAE mice was associated with reduced infiltration of encephalitogenic IL17A and GM-CSF producing donor-derived T helper cells and recipient-derived inflammatory myeloid cells in the brain and spinal cord. In addition, we found that in vitro PTX-treated donor Th17 cells do not sufficiently accumulate in the CNS and have a reduced capacity to induce EAE. Notably, there were reduced numbers of neutrophils in the CNS of mice that received in vitro PTX-treated Th17 cells. GM-CSF produced by the Th17 cells is essential for EAE induction [31,40] and IL17A contributes to the development of disease [41]. GM-CSF directly activates inflammatory macrophages and promotes the production of macrophage-tropic chemokines, such as CCL6, CCL24 and CCL17, which drives inflammatory processes during disease [42,43,44]. Although neutrophils are one of the major GM-CSF responsive cell types, IL17A produced by the T cells has been shown to promote neutrophil recruitment in the brain [45]. Taking all this into consideration, we attribute the ameliorated disease in PTX-treated Th17-EAE mice to the restricted accumulation of GM-CSF^+^ and, to a lesser extent, IL-17A^+^, CD4^+^ T helper cells in the CNS.

The chemokine receptors, CCR2 and CCR6, are utilized by Th17 cells to enter into the CNS during EAE. Experimental evidence demonstrates that the initial trigger of inflammation is caused by CCR6-dependent infiltration of autoreactive Th17 cells into the uninflamed CNS, which is then followed by a second wave of CCR2-dependent recruitment of highly pathogenic GM-CSF^+^ and/or IFNγ^+^ Th17 cells [33,46]. It has been suggested that persistent engagement with antigens at this stage of the disease promotes the development of these encephalitogenic CCR2^+^ “ex-Th17” cells in a Tbet-dependent manner [33]. As previously mentioned, PTX has been known to compromise the chemo-attraction of GPCR-expressing cells by its ADP-ribosyltransferase activity [19,47]. In this study, we observed that Th17 cells polarized in the presence of PTX were significantly less responsive to CCR2-CCL2- and CCR6-CCL20-driven chemotaxis compared to the control. This finding supports our hypothesis that PTX treatment directly affects Th17 cells, by inhibiting their recruitment to the CNS, to inhibit disease. However, both IL17A^+^-producing Th17 and IFNγ^+^-producing Th1 cells are able to migrate into the CNS using GPCR-dependent chemotaxis. Therefore, in addition to inhibiting the migration of Th17 cells, it is possible that PTX also inhibits Th1 cell migration. As the donor T cells were enriched with IL-23, and only CCL2-CCR2-dependent trafficking of IFNγ^+^ cells was compromised by PTX, we hypothesize that the IFNγ^+^ cells are “ex-Th17” cells rather than “classical” Th1. Further investigation is required to delineate the effects of PTX on the “ex-Th17” and “classical” Th1 cells in the EAE model. In contrast to the inhibitory effects on T cells and myeloid cells, we found that PTX promoted B-cell accumulation in the CNS. As B cells can have either inflammatory or regulatory functions [48], it was initially unclear what function B cells had in the Th17-EAE mice treated with PTX. Here, we find that PTX treatment of B-cell-deficient (µMT) mice during Th17-EAE significantly diminished EAE severity and reduced the numbers of Th17 cells in the CNS compared to the WT-PTX-treated mice, which demonstrates an inflammatory role for B cells. However, the differences between WT and µMT were not apparent when PTX was not administered to mice. Interestingly, we found that the elevated B cell numbers in the CNS of mice were not a direct function of PTX. We assessed the immune cell populations in the No PTX mice at two time points based on the severity of disease (early: severe disease; late: mild disease). We found that there was an increase in the proportion of accumulated B cells in the spinal cord of the mice with a milder disease, which were therefore sacrificed at a later time point, compared to the mice that had severe disease. Taken together, these observations suggest that PTX allows for the accumulation of B cells by suppressing the infiltration of cytokine-producing T helper cells and myeloid cells into the CNS. Furthermore, by inhibiting the infiltration of inflammatory myeloid cells, B cells become critical for the development of disease.

We currently do not know the precise reason that PTX treatment promotes a B-cell-driven disease. However, we found that PTX treatment increased the number of MHCII^+^ B cells in the CNS and CD80 and CD86 expression in the splenic B cells. These data suggest that PTX increases the capacity of B cells to present antigens to T cells, which is a required pathogenic mechanism in other B-cell-dependent EAE models [5,6]. It has been shown that the resting peripheral B cells are not able to activate Th17 cells to produce IL17A in vitro, unless there is exogenous IL1β present [10]. Therefore, it is also possible that PTX-induced IL1β elevates the capacity of peripheral B cells to activate Th17 cells to drive disease.

Due to the efficacy of B-cell-depleting therapies using Rituximab [1,2] and Ocrelizumab [3], defining the mechanism by which B cells contribute to neuroinflammation in MS is of high interest. Since anti-CD20 treatment does not deplete antibody-producing plasma cells, it indicates an alternate pathogenic function of B cells during MS which is not completely understood. Several studies have established the importance of B–T cell cross talk during MS and EAE [11,12,13]. Our study provides more evidence that B cells play an important role in Th17-driven neuroinflammation in mice. This is a clinically relevant immunological mechanism because B-cell-depleting therapies diminish inflammatory Th17 responses in MS patients [49].

In conclusion, we systematically show that PTX functionally inhibits the CNS infiltration of encephalitogenic T cells, while promoting an inflammatory B-cell phenotype in the periphery and allowing the accumulation of pathogenic B cells into the CNS. Whether or not the pathogenic function of B cells in this model is due to its antigen-presenting function will require further investigation. This demonstrates that PTX alters the immunopathology of Th17-EAE toward a B-cell-dependent model, which will serve as an important tool to study the pathogenic mechanisms of B cells in MS.

## 4. Materials and Methods

### 4.1. Mice

Eight- to ten-week-old female C57BL/6 (WT, CD45.2^+^), B6.SJL-Ptprca Pepcb/BoyJ (CD45.1^+^) and B6.129S2-Ighmtm1Cgn/J (μMT) mice were purchased from Jackson Laboratory and housed in the Oklahoma Medical Research Foundation animal facility. All animals were housed and treated in compliance with the institutional IACUC.

### 4.2. EAE Induction

For adoptive transfer Th17-EAE, (C57BL/6) or CD45.1^+^ donor mice were immunized with 150 µg MOG_35–55_ peptide (Genemed Synthesis Inc., San Antonio, TX, USA) emulsified in complete Freund’s adjuvant (5 mg/mL heat-killed M. tuberculosis), followed by intraperitoneal injections (IP) of 250 ng of PTX (List Biological Laboratories, Inc., Campbell, CA, USA) in 200 µL of PBS on day 0 and 2 post-immunization. At day 10, the spleens and lymph nodes were harvested, mechanically disrupted and a single-cell suspension was generated. For Th17 polarization, the cells were cultured at 2.5 × 10^6^/ml concentration in complete RPMI 1640 with 10 µg/mL MOG_35–55_ peptide, 10 ng/mL IL-23 (R&D systems, Minneapolis, MN, USA) and 10 µg/mL IFN-γ antibody (InVivoMAb, Bio X Cell, Lebanon, NH, USA) for 72 h at 37 °C. Then, 1.5 × 10^7^ cells in 200 µL PBS were transferred (IP) into WT or µMT recipient mice. For the in vivo PTX treatment, the recipient mice received 250 ng PTX in 200 µL of PBS (IP) at day 0 and 2 post-transfer. The control animals (No PTX) received 200 µL of PBS. For in vitro PTX experiments, 100 ng/mL PTX was added to the Th17 polarization media.

Paralysis was monitored daily using a standard clinical score: (1) loss of tail tone, (2) incomplete hind limb paralysis, (3) complete hind limb paralysis, (4) forelimb paralysis and (5) moribund/dead.

A Kaplan-Meier survival curve was generated to determine the experimental duration (day 17) for Th17-EAE mice that were not treated with PTX (No PTX). The experimental duration for all other groups was determined when the disease severity of the entire group was at a maximum (peak of disease, in vivo PTX groups: day 30–35; in vitro PTX groups: day 27,28).

### 4.3. Isolation of CNS-Infiltrating Cells

Animals were sacrificed at the peak of disease and cells were isolated from the brain (cerebellum) and spinal cord of mice perfused with PBS. Brain and spinal cord tissue was mechanically disrupted and incubated with 5 µL/mL DNAse (Sigma, St. Louis, MO, USA) and 4 mg/ml collagenase (Roche, San Francisco, CA, USA) for 40 minutes at 37 °C. Post-incubation, the tissues were homogenized mechanically and cells were purified from myelin using a 40% Percoll (GE healthcare) gradient.

### 4.4. Flow Cytometry

The following antibodies were used for flow cytometry: αCD4-APC/PE-Cy7/BV605 (GK1.5); αCD44-APC/PE-Cy7 (IM7); αCD19-AF488/APC-Cy7 (6D5); αB220-PE (RA3-6B2); αCD138-BV605 (281-2); αIgD-PE/APC-Cy7 (11-26c.2a); αIgM-e450 (II-41), αCD38-PE (90); αCD11b-PerCPCy5.5/e450(M1/70), αMHCII-BV711(M5/114.15.2), αLy6G-AF647(A18); αF4/80-Pacific blue (BM8); αCD45.1-BV785 (A20); αCD45.2-BV605 (104) (Biolegend, San Diego, CA, USA). Fc-block was used prior to surface staining. For testing viability, Propidium Iodide or Zombie Red (Biolegend) were used.

For intracellular cytokine staining, cells were first stimulated with phorbol 12-myristate 13-acetate (PMA) (50 ng/ml, Sigma-Aldrich, St. Louis, MO, USA), ionomycin (500 ng/ml, Sigma-Aldrich, St. Louis, MO, USA) and Golgi Stop (BD Biosciences, Franklin Lakes, NJ, USA) for 3.5 hours at 37 °C. Cells were stained with fixable viability dye Zombie Red (Biolegend, San Diego, CA, USA). Cells were stained with appropriate surface antibodies listed above. Cells were then fixed and permeabilized with Cytofix/Cytoperm (BD Biosciences) before staining with αIL-17A-FITC (TC11-18H10.1), αIL10-PE/PE-Cy7 (JES5-16E3), αGM-CSF-PerCPCy5.5 (MP1-22E9) and αIFNγ-BV711 (XMG1.2) (Biolegend). All flow cytometry data were acquired with BD LSRII and analyzed with FlowJo_V10 software (Tree Star Inc., Ashland, OR, USA).

### 4.5. Histology

Fixed (4% paraformaldehyde in PBS) tissue sections of brain and spinal cord were paraffin-embedded, cut and stained with H&E and Luxol Fast Blue, according to standard protocols.

### 4.6. Chemotaxis Assay

Chemotaxis assay was performed in 24-well plates using 6.5-mm inserts with 5-µm-pore-size polycarbonate membranes (Corning, Corning, NY, USA). Cells were isolated from spleens and lymph nodes of MOG_35-55_/ complete Freund’s adjuvant (CFA) immunized mice at day 10 and polarized using Th17 culture media (as described above) in the presence or absence of PTX (100 ng/ml) for 72 hours at 37 °C. Post-culture, cells were washed with PBS and re-suspended in pre-warmed serum-free complete RPMI 1640. Pre-warmed complete RPMI 1640 (600 µL) containing chemokine CCL2 (1 ng/ml and 5 ng/ml; R&D Systems) or CCL20 (10 ng/ml and 50 ng/ml; R&D Systems) was added in the lower chamber. Then, 2 × 10^6−^cells in 100 µL of serum-free RPMI 1640 were added to the upper chamber of the trans-well inserts and incubated for 5 hours at 37 °C. At the end of incubation, the cells from the lower chamber were collected and absolute numbers of migrated cells were determined using Countess II FL Automated Cell Counter (Thermo-fisher Scientific, Waltham, MA, USA). Cells were prepared and stained for flow cytometry as described above. Migration index was calculated by normalizing the number of cells in the lower chamber to the number of cells in the lower chamber of the “no-chemokine” control.

### 4.7. Statistics

Data are presented as mean ± SEM and statistical significance was determined using a one-way ANOVA, two-tailed Student’s *t*-test or Mann-Whitney test. Log-rank (Mantel-Cox) test was used to determine statistical significance between survival curves. All statistical analyses were performed using Prism 7 (GraphPad, San Diego, CA, USA). Differences between two datasets were considered statistically significant if *p* value was less than 0.05.

## Figures and Tables

**Figure 1 ijms-22-02924-f001:**
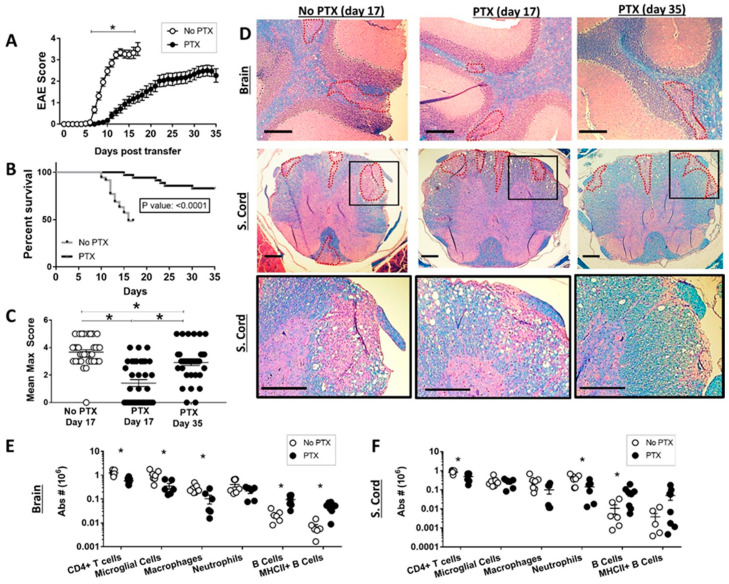
Pertussis toxin ameliorates disease in the Th17-EAE model. Myelin oligodendrocyte glycoprotein (MOG)-specific Th17 cells were transferred into recipient mice and treated with 250 ng pertussis toxin (PTX) intraperitoneally at day 0 and 2 post-transfer of cells. (**A**) Clinical scores of mice with Th17-EAE treated with PTX or PBS (No PTX). (**B**) Kaplan-Meier survival curve of Th-17 EAE mice treated with (PTX) or without (No PTX) at day 0 and 2 post-transfer. (**C**) Mean maximum score of mice treated with PTX or PBS (No PTX); *n* = 30 mice per group. Mann-Whitney tests were performed to determine statistical significance (* *p* < 0.05). Results compiled from 5 independent experiments. (**D**) Representative brain and spinal cord sections from No PTX (day 17) and PTX groups (day 17 and day 35) with similar EAE scores and stained with H&E and Luxol Fast Blue; scale bar 200 µm. Fewer and smaller demyelinated lesions (regions enclosed with red dotted lines) were observed in the PTX group. Absolute numbers of viable immune cells in the (**E**) brain and (**F**) spinal cord (No PTX day 17 and PTX day 35) were determined by flow cytometry; CD4^+^ T cells, microglial cells (CD45^int^CDllb^int^), macrophages (CD45^hi^CDllb^hi^MHCII^+^ly6G^−^), neutrophils (CD45^hi^CDllb^hi^MHCII^−^ly6G^+^) and B cells (CD19^+^B220^+^). Data collected from 2 independent experiments; *n* = 6–8/group. Statistical analysis was performed using Mann-Whitney tests (* *p* < 0.05).

**Figure 2 ijms-22-02924-f002:**
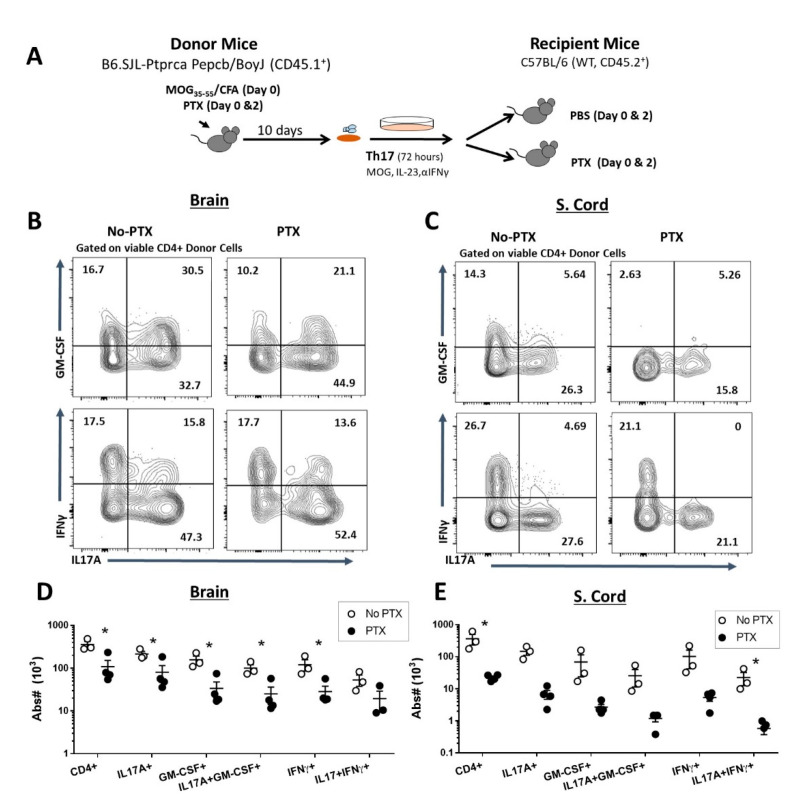
Pertussis toxin reduced accumulation of encephalitogenic CD4^+^ T cells in the CNS. (**A**) Th17-EAE was induced in wild-type CD45.2^+^ mice with Th17 cells derived from CD45.1^+^ donor mice and treated with PTX (250 ng IP) or PBS (No PTX) on day 0 and 2 post-transfer. Representative flow cytometry plots of IL17A^+^GM-CSF^+^ and IL17+IFNγ expressing CD4^+^ donor T cells in the (**B**) brain and (**C**) spinal cord of mice with Th17-EAE treated with or without PTX. Gated on viable CD4^+^CD45.1^+^ population in the brain and spinal cords of mice with Th17-EAE. Absolute number of IL17A^+^, GM-CSF^+^, IFNγ^+^, IL17A^+^GM-CSF^+^ and IL17A^+^IFNγ^+^ CD4^+^CD45.1^+^ donor T cells in the (**D**) brain and (**E**) spinal cord of mice with Th17-EAE treated with (PTX) or PBS (No PTX); *n* = 3–4/group. Statistical analysis was performed using Student’s *t-*test (* *p* < 0.05).

**Figure 3 ijms-22-02924-f003:**
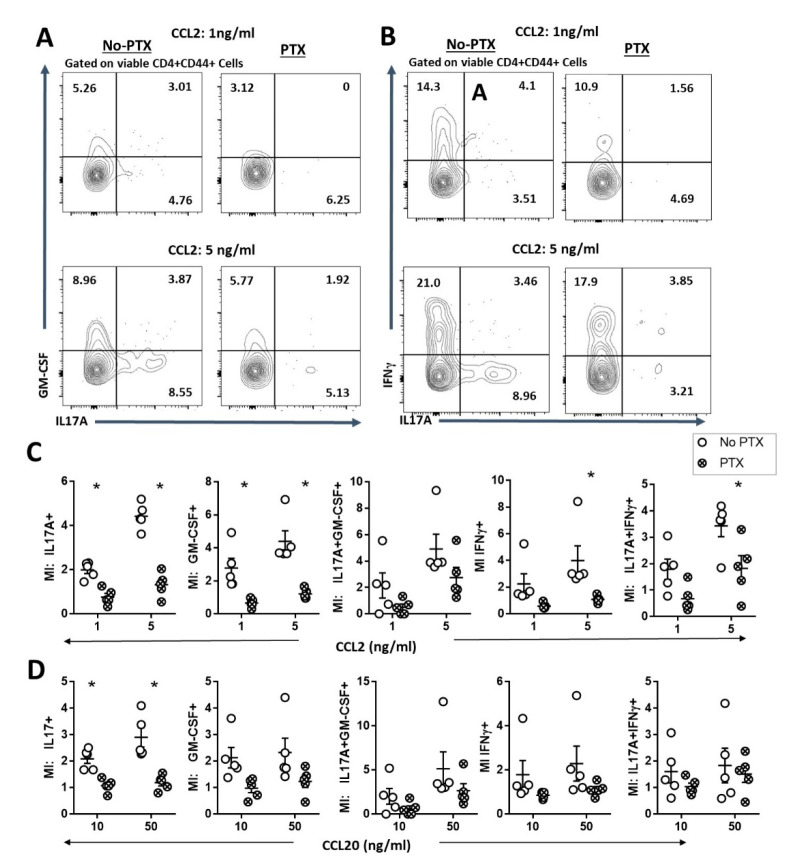
Pertussis toxin inhibited chemotaxis of activated Th17 cells in culture. Trans-well chemotaxis to CCL2 and CCL20 was performed on cells harvested from spleens and lymph nodes of MOG/CFA immunized mice at day 10 and polarized under Th17 conditions with or without PTX (100 ng/ml) for 72 hours. (**A**) Representative flow cytometry plots of migrated CD4^+^CD44^+^ T cells expressing IL17A and GM-CSF in the lower chamber. (**B**) Representative flow cytometry plots of migrated CD4^+^CD44^+^ T cells expressing IL17A and IFNγ in the lower chamber. (**C**) Migration index (MI) to CCL2-dependent chemotaxis of IL17A^+^, GM-CSF^+^, IFNγ^+^, IL17A^+^GM-CSF^+^ and IL17A^+^IFNγ^+^ CD4^+^CD44^+^ T cells in the lower chamber. (**D**) Migration index (MI) to CCL20-dependent chemotaxis of IL17A^+^, GM-CSF^+^, IFNγ^+^, IL17A^+^GM-CSF^+^ and IL17A^+^IFNγ^+^ CD4^+^CD44^+^ T cells in the lower chamber. Migration index was calculated by normalizing the absolute number of cells to the no-chemokine control; *n* = 5/group. Statistical analysis was performed using Student’s *t*-test (* *p* < 0.05).

**Figure 4 ijms-22-02924-f004:**
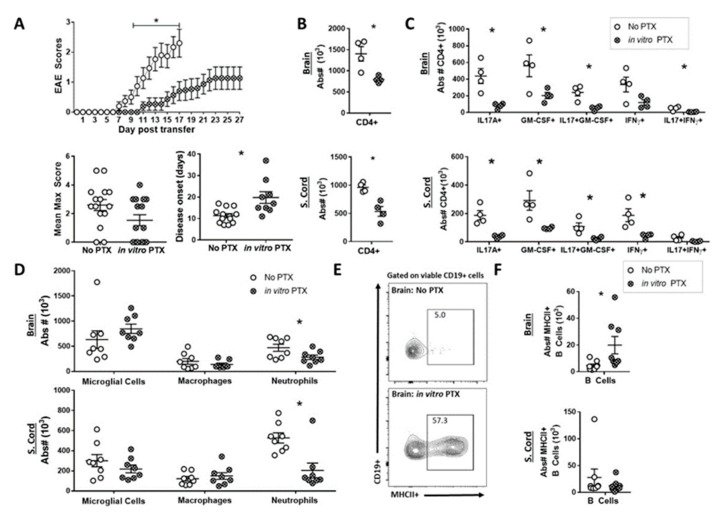
Th17 cells polarized with PTX induced weaker disease during Th17-EAE. MOG-specific Th17 cells were polarized in the presence of 100 ng/mL PTX for 72 hours in vitro and transferred into healthy recipient mice. (**A**) Clinical scores, mean maximum score and disease onset of mice receiving MOG-specific Th17 cells cultured with (in vitro PTX) or without PTX (No PTX); *n* = 15/group. Mann-Whitney tests were performed to determine statistical significance (* *p* < 0.05). Results compiled from 3 independent experiments. (**B**) Absolute number of CD4^+^ T cells in the brain and spinal cord; *n* = 4/group; Mann-Whitney tests were performed to determine statistical significance (* *p* < 0.05). Data representative of 2 independent experiments. (**C**) Absolute numbers of viable IL17A^+^, GM-CSF^+^, IFNγ^+^, IL17A^+^GM-CSF^+^ and IL17A^+^IFNγ^+^ CD4^+^ T cells in the brain and spinal cords of Th17-EAE mice; *n* = 4/group; Mann–Whitney tests were performed to determine statistical significance (* *p* < 0.05). Data representative of 2 independent experiments. (**D**) Absolute number of microglial cells (CD45^int^CDllb^int^), macrophages (CD45^hi^CDllb^hi^MHCII^+^ly6G^−^) and neutrophils (CD45^hi^CDllb^hi^MHCII^-^ly6G^+^) in the brain and spinal cord of Th17-EAE mice; *n* = 8/group; results compiled from 2 independent experiments. Statistical analysis was performed using Student’s *t-*test (* *p* < 0.05). (**E**) Representative flow cytometry plots of viable CD19^+^MHCII^+^ cells in the brain of Th17-EAE mice. (**F**) Absolute numbers of CD19^+^MHCII^+^ cells in the brain and spinal cord of Th17-EAE mice; *n* = 8/group; results compiled from 2 independent experiments. CNS-infiltrating cells were assessed on day 17 for the no PTX mice and day 27 for the in vitro PTX mice. Statistical analysis was performed using Student’s *t*-test (* *p* < 0.05).

**Figure 5 ijms-22-02924-f005:**
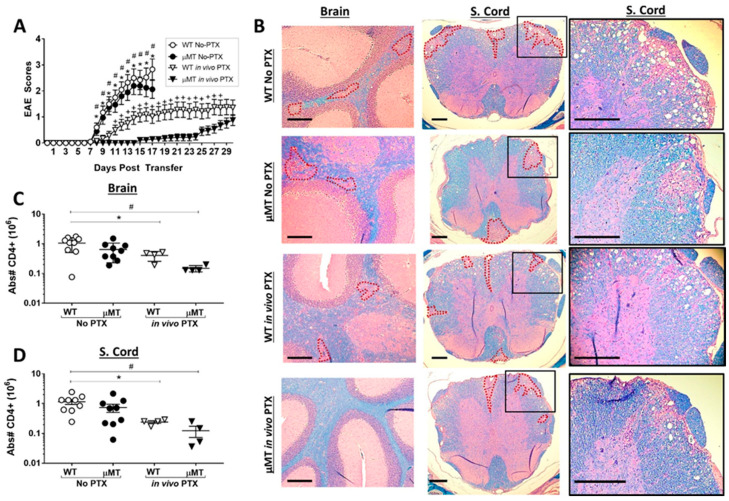
B-cell-deficient mice were protected against Th17-EAE when treated with PTX. (**A**) Clinical scores of WT and B-cell-deficient (µMT) mice with Th17-EAE treated with PTX (250 ng IP) (in vivo PTX) or PBS (No PTX) on day 0 and 2 post-transfer; *n* = 25–30/group; results compiled from 5 independent experiments. Mann–Whitney tests were performed between each group to determine statistical significance (* *p* < 0.05, WT No PTX vs. WT in vivo PTX; ^#^
*p* < 0.05 WT No PTX vs. µMT in vivo PTX; ^+^
*p* < 0.05, WT in vivo PTX vs. µMT in vivo PTX). (**B**) Representative brain and spinal cord sections of WT and µMT mice stained with H&E and Luxol Fast Blue (regions enclosed with red dotted lines); scale bar 200 µm. Absolute number of CD4^+^ T cells in the (**C**) brain and (**D**) spinal cord of WT and µMT mice with PTX or PBS treatment; *n* = 8/group. Data representative of 2 independent experiments. One-way ANOVA was performed to determine statistical significance (* *p* < 0.05, WT No PTX vs. WT in vivo PTX; ^#^
*p* < 0.05 WT No PTX vs. µMT in vivo PTX).

**Figure 6 ijms-22-02924-f006:**
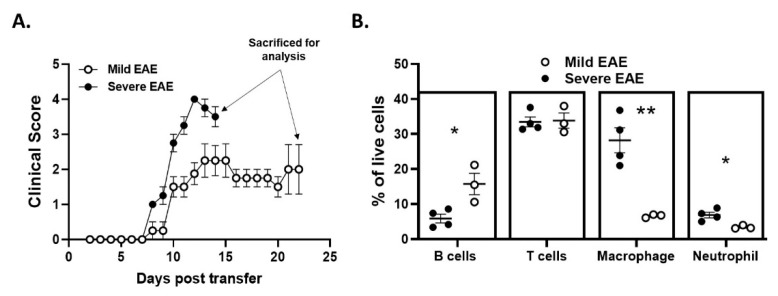
Disease duration and/or severity affects B-cell and myeloid cell accumulation into the CNS of mice. Recipient mice received donor Th17 cells and were not treated with PTX. Severe EAE: sacrificed on day 14, *n* = 4/group; Mild EAE: sacrificed on day 22, *n* = 3/group. (**A**) EAE scores were monitored in mice with mild and severe EAE. (**B**) Comparison of the percentage of B cells, T cells, macrophages and neutrophils in the spinal cords was assessed by flow cytometry in the mild and severe EAE mice on the day of sacrifice. Statistical differences were found using Student’s *t*-tests. * *p* < 0.05 and ** *p* < 0.01. B cells: CD19^+^; T cells: CD4^+^; Macrophages: CD11b^+^F4/80^+^; Neutrophils: CD11b^+^Ly6G^+^.

## Data Availability

The data presented in this study are available on request from the corresponding author.

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
