# Peer review of "Pertussis Toxin Inhibits Encephalitogenic T-Cell Infiltration and Promotes a B-Cell-Driven Disease during Th17-EAE"

_ijms, 2021, doi:10.3390/ijms22062924_

Round 1

Reviewer 1 Report

In this work, Maria et al show that pertussis toxin (PTX), that is necessary for the induction of active EAE, exerts a very different effect in the passive EAE model and changes the disease from a Th17-driven model to an encephalitogenic B cell-driven model, serving as a new approach that evidences the important role of B cells in multiple sclerosis and open the field for a better understanding of the disease.

Technically, the experiments are well done and explained.

Major comments:

  • Regarding Figure 1A, authors explain that there is a delay in the onset of the disease and that analysis was done in the peak of the disease. Could authors specify the day, and if both groups (-/+ PTX) were sacrificed the same day?
  • In figure 1A, 4A and 5A, scores for No PTX group end at day 17. Is there a reason why these groups do not reach the end of the experiment at day 27/35? If animals died/were sacrificed due to human criteria, a Kaplan-Meier survival curve is necessary. If this is not the case, could authors refer the EAE scores for these animals at day 27/35? It is highly recommended in order to see if this effect affects only the onset of the disease or ameliorates the phenotype.
  • Regarding the representative pictures in Figure 1, it is not clear at which timepoint these pictures were taken. In addition, and given that this is a disease with accumulative damage, the representative images that should be used are the ones at the end of the experiment, very often EAE scores can be slightly subjective. Furthermore, authors should show a magnification of the areas of interest, with the current magnification is difficult to see the results explained by the authors, and the use of dotted lines for delimiting the demyelinated area would help to emphasize authors’ statements.
  • Lines 151-155: there is a mistake referring to supplemental figure 1. CD80, CD86 and MHCII are evaluated in supplemental figure 2, not 1.
  • It is appreciated the use of a cartoon in figure 2 to explain the experimental design.
  • In section 3.2, please specify that the reduction in IL-17+ cells in the brain is related to absolute numbers, the representative plots in figure 2B have shown an increase in the proportion.
  • In figure 5, demyelination is still evident in uMT mice with PTX treatment. Please, emphasize what authors consider lesions and clarify it in the text (only demyelinated areas, demyelination+presence of infiltrated cells, etc)

Minor comments:

  • Authors need to be consistent in the nomenclature: IFNγ, not IFNg; in vitro and in vivo in italic, etc.

Author Response

Reviewer 1:

Point 1: “Regarding Figure 1A, authors explain that there is a delay in the onset of the disease and that analysis was done in the peak of the disease. Could authors specify the day, and if both groups (-/+ PTX) were sacrificed the same day?

In figure 1A, 4A and 5A, scores for No PTX group end at day 17. Is there a reason why these groups do not reach the end of the experiment at day 27/35? If animals died/were sacrificed due to human criteria, a Kaplan-Meier survival curve is necessary. If this is not the case, could authors refer the EAE scores for these animals at day 27/35? It is highly recommended in order to see if this effect affects only the onset of the disease or ameliorates the phenotype.”

Response: In our experiments, the No-PTX treated group had severe disease where about 50% of the mice were reached a score greater than 4 or had lost over 30% of body weight that they needed to be euthanized to abide by our IACUC protocol. As you have pointed out, we have now perform a Kaplan Meyer test to compare PTX to no PTX in Figure 1B. (Lines 86-90)

Point 2: “Regarding the representative pictures in Figure 1, it is not clear at which timepoint these pictures were taken. In addition, and given that this is a disease with accumulative damage, the representative images that should be used are the ones at the end of the experiment, very often EAE scores can be slightly subjective. Furthermore, authors should show a magnification of the areas of interest, with the current magnification is difficult to see the results explained by the authors, and the use of dotted lines for delimiting the demyelinated area would help to emphasize authors’ statements.”

Response: We have provided histology from mice at day 17 (No PTX and PTX treated mice) as well as day 35 (PTX treated) in Figure 1D. We have also outlined the lesions with a red dotted line and show magnified pictures of the lesion areas.

Point 3: “Lines 151-155: there is a mistake referring to supplemental figure 1. CD80, CD86 and MHCII are evaluated in supplemental figure 2, not 1.”

Response: We have corrected these errors. (Lines 174-176))

Point 4: “In section 3.2, please specify that the reduction in IL-17+ cells in the brain is related to absolute numbers, the representative plots in figure 2B have shown an increase in the proportion.”

Response: We have now specified that this is a reduction in the absolute number of Th17 cells in the brain. (Lines 205-211))

Point 5: “In figure 5, demyelination is still evident in uMT mice with PTX treatment. Please, emphasize what authors consider lesions and clarify it in the text (only demyelinated areas, demyelination+presence of infiltrated cells, etc).”

Response: We have now described the presence of lesions in the spinal cords of the PTX-treated µMT mice. (Lines 300-301)

Point 6: Authors need to be consistent in the nomenclature: IFNγ, not IFNg; in vitro and in vivo in italic, etc.

Response: We have corrected these typos.

Reviewer 2 Report

The article by Maria and colleagues describes the effect of Pertussis Toxin treatment on Th17 adoptive transfer EAE. The topic is of high interest as the mechanisms of action of pertussis toxin on the immune system are complex and can significantly contribute to our understanding of the cellular mechanisms of pathogenesis. The authors show that PTX inhibits Th17-driven disease and suggest that PTX promotes B cell pathogenicity. The drawn conclusions are intriguing, however, in some instances premature/based on incomplete datasets. Thus, the following points need to be addressed to support the conclusions:

  1. It is not at all a novel observation that PTX selectively inhibits Th17 cell migration. In fact, the first anecdotal evidence is from 1996 when Robbinson and colleagues showed that PLP-specific T cells lose their encephalitogenic potential upon adoptive transfer when cultured with PTX (PMID 8640862). Furthermore, it was already demonstrated that primary MOG-specific Th17 cells do not induce EAE when recipients are treated with PTX (Jäger et al., 2009 JI). While the authors investigate the phenomenon in more detail, it is not adequate to portray this as their own novel observation and the earlier studies should be discussed and cited.
  2. The authors are claiming that specifically Th17 cells are inhibited by PTX. They show good evidence for this, however, the evidence for Th1 cells not being affected by PTX is actually limited. In order to claim Th17-specificity some questions should be answered:
  • The transferred Th17 cells are not pure, since they are derived from immunized animals and enriched for Th17 cells via culture in presence of IL-23. What is the cytokine profile of these cells at time of transfer, how much IFNg and Tbet do they express and how much IL-17? If the data in Fig. S4 is representative they produce very little cytokine and equally low amounts of IL-17 and IFNg?
  • The migration assay is performed with ligands for the Th17-associated receptors CCR2 and CCR6 – are these the dominant receptors expressed on the cells at the time of transfer? What about (the more Th1-associated) CXCR3, is it expressed and if so, is migration towards its ligands also inhibited by PTX? Also, for the migration assay it would be good to show the cytokine profile including IFNg before migration with and without PTX.
  • In Fig. S3 the authors show that PTX-treated recipients have not only reduced IL-17 and GM-CSF-positive CD4 T cells in the CNS but also reduced IFNg-positive T cells. However, in the congenic transfer experiment (Fig.2) no IFNg-data is shown – do the transferred cells produce IFNg in the CNS?

3. The authors observe that PTX-treated animals develop disease later but show a (mild) increase of B cell infiltration into the CNS and conclude that ‘PTX promotes the accumulation of B cells in the CNS’. This is a premature conclusion because the mice were analyzed at very different times (according to M&M): the control group at peak of disease probably around 12 days post transfer and the PTX-treated group at peak probably 25 days or even later after T cell transfer? This means that B cells had at least 10 more days in the PTX-treated group to be recruited to the CNS and up-regulate MHC II and thus, we may simply look at a function of time rather than a PTX-dependent effect here. At least in the transfer experiment with in vitro-PTX-treated Th17 cells (Fig. 4) the disease course in the control group seems mild enough that it should be feasible to analyze some animals at the same time point after T cell transfer to address whether PTX really increases B cell recruitment and upregulation of MHC II.

4. The authors suggest in the discussion that PTX directly acts on B cells and ‘increases the capacity of B cells to present antigen to T cells’, however, their own data (Fig.4) shows that in vitro-PTX treated Th17 cells recruit more MHC II-positive B cells to the CNS. If the time of sampling can be ruled out as a confounding factor, this would argue rather for an indirect effect of PTX on B cell recruitment mainly via manipulation of the T cells, no? Here the discussion should be refined..

Author Response

Reviewer 2:

Point 1: “It is not at all a novel observation that PTX selectively inhibits Th17 cell migration. In fact, the first anecdotal evidence is from 1996 when Robbinson and colleagues showed that PLP-specific T cells lose their encephalitogenic potential upon adoptive transfer when cultured with PTX (PMID 8640862). Furthermore, it was already demonstrated that primary MOG-specific Th17 cells do not induce EAE when recipients are treated with PTX (Jäger et al., 2009 JI). While the authors investigate the phenomenon in more detail, it is not adequate to portray this as their own novel observation and the earlier studies should be discussed and cited.”

Response: We thank the reviewer for pointing out these key papers.  We have now cited them appropriately in the introduction (lines 54-56) and the discussion (lines 360-369).

Point 2. “The authors are claiming that specifically Th17 cells are inhibited by PTX. They show good evidence for this, however, the evidence for Th1 cells not being affected by PTX is actually limited. In order to claim Th17-specificity some questions should be answered:

 The transferred Th17 cells are not pure, since they are derived from immunized animals and enriched for Th17 cells via culture in presence of IL-23. What is the cytokine profile of these cells at time of transfer, how much IFNγ and Tbet do they express and how much IL-17? If the data in Fig. S4 is representative they produce very little cytokine and equally low amounts of IL-17 and IFNγ?

Response: We have revised this figure to show the percentages of cytokine producing cells in the activated (CD44+) CD4+ gate (Fig S4A-B) as well as the percentages of the cytokine producing CD44+CD4+ cell of the live gate (Fig S4C).  These show that the activated cells (presumably mostly MOG specific) are producing a substantial amount of cytokines including IL17, GM-CSF and IFNγ (Fig S4A-B).  (Note: We do not see much cytokine by the CD44-CD4+ cells). Pertussis toxin does not affect the expression of these cytokines. We did not assess Tbet expression in these cells.  We had one week to revise the manuscript, which was not enough time to perform this experiment.  We found that in vitro PTX treatment did not alter IFNγ expression, which demonstrates that PTX has no overt effects on the “Th1” or “ex-Th17” pathways.

Point 3: “The migration assay is performed with ligands for the Th17-associated receptors CCR2 and CCR6 – are these the dominant receptors expressed on the cells at the time of transfer? What about (the more Th1-associated) CXCR3, is it expressed and if so, is migration towards its ligands also inhibited by PTX? Also, for the migration assay it would be good to show the cytokine profile including IFNγ before migration with and without PTX.”

Response: We did not address this question due to the short turn around for submitting revisions. We feel that the lack of CXCR3 trafficking does not detract from the central findings of this manuscript. The cytokine profile of the cells prior to the migration assay are included in Figure S4A-D.

Point 4: In Fig. S3 the authors show that PTX-treated recipients have not only reduced IL-17 and GM-CSF-positive CD4 T cells in the CNS but also reduced IFNγ-positive T cells. However, in the congenic transfer experiment (Fig.2) no IFNg-data is shown – do the transferred cells produce IFNγ in the CNS?

Response: Yes, the transferred donor CD45.1 donor cells do produce IFNγ and their infiltration is inhibited by PTX.  This is depicted in Figure 2B-E.

Point 5. “The authors observe that PTX-treated animals develop disease later but show a (mild) increase of B cell infiltration into the CNS and conclude that ‘PTX promotes the accumulation of B cells in the CNS’. This is a premature conclusion because the mice were analyzed at very different times (according to M&M): the control group at peak of disease probably around 12 days post transfer and the PTX-treated group at peak probably 25 days or even later after T cell transfer? This means that B cells had at least 10 more days in the PTX-treated group to be recruited to the CNS and up-regulate MHC II and thus, we may simply look at a function of time rather than a PTX-dependent effect here. At least in the transfer experiment with in vitro-PTX-treated Th17 cells (Fig. 4) the disease course in the control group seems mild enough that it should be feasible to analyze some animals at the same time point after T cell transfer to address whether PTX really increases B cell recruitment and upregulation of MHC II.”

Response: This is a great point.  The attenuation of the early inflammation, which is myeloid cell dominant, may actually reveal the importance of a pathogenic B cell response in these mice.  We have now analyzed mice with mild Th17-induced disease that were sacrificed at a later time point. We found that that there was an accumulation of B cells in the spinal cords at this time point.  We have added these data to the manuscript (Figure 6) (lines 318-337) and have added these valid points to the discussion (lines 415-423). 

Point 6: The authors suggest in the discussion that PTX directly acts on B cells and ‘increases the capacity of B cells to present antigen to T cells’, however, their own data (Fig.4) shows that in vitro-PTX treated Th17 cells recruit more MHC II-positive B cells to the CNS. If the time of sampling can be ruled out as a confounding factor, this would argue rather for an indirect effect of PTX on B cell recruitment mainly via manipulation of the T cells, no? Here the discussion should be refined.

Response: See response to point 5. 

Round 2

Reviewer 2 Report

The discussion of previous work is now adequate. Point 1-2 are well addressed.

Quite some data has been added regarding the expression of IFNg by the transferred Th17 cells and the text and discussion have been refined to reflect that also IFNg-producing T cells are affected in their migration in response to PTX. A CXCR3-dependent migration assay would have been nice, but understandably that cannot be performed in one week.

Point 5 and 6 have been addressed and it is now much clearer that B cell accumulation increases mainly due to the longer disease course in PTX treated mice. Still for the timing of analysis it should be pointed out in the legends if the mice were analyzed at very different time points (for example in Fig. 1E/F, d17 for ctrl and d35 for PTX-group (?), also in Fig. 4 and S4).

Author Response

Thank you for your comments.

We have added the days of analysis to Figure 1, Figure 4 and Supplemental Figure 4.  These new edits are highlighted in green.